# Psychometric Performance of HRQoL Measures: An Australian Paediatric Multi-Instrument Comparison Study Protocol (P-MIC)

**DOI:** 10.3390/children8080714

**Published:** 2021-08-20

**Authors:** Renee Jones, Brendan Mulhern, Kristy McGregor, Shilana Yip, Rachel O’Loughlin, Nancy Devlin, Harriet Hiscock, Kim Dalziel

**Affiliations:** 1Health Services, Murdoch Children’s Research Institute, Melbourne 3052, Australia; renee.jones@mcri.edu.au (R.J.); kristy.mcgregor@mcri.edu.au (K.M.); shilana.yip@mcri.edu.au (S.Y.); Harriet.Hiscock@rch.org.au (H.H.); 2Centre for Health Economics Research and Evaluation, University of Technology Sydney, Ultimo 2007, Australia; Brendan.Mulhern@chere.uts.edu.au; 3Centre for Health Policy, Health Economics Unit, The University of Melbourne, Melbourne 3053, Australia; rneely@student.unimelb.edu.au (R.O.); nancy.devlin@unimelb.edu.au (N.D.); 4Health Services Research Unit, Royal Children’s Hospital, Melbourne 3052, Australia; 5Department of Paediatrics, The University of Melbourne, Melbourne 3010, Australia

**Keywords:** pediatrics, quality of life, cost-benefit analysis, psychometrics, chronic disease, mental health, patient reported outcome measures

## Abstract

Background: There is a lack of psychometric evidence about pediatric health-related quality of life (HRQoL) instruments. Evidence on cost effectiveness, involving the measurement of HRQoL, is used in many countries to make decisions about pharmaceuticals, technologies, and health services for children. Additionally, valid instruments are required to facilitate accurate outcome measurement and clinical decision making. A pediatric multi instrument comparison (P-MIC) study is planned to compare the psychometric performance and measurement characteristics of pediatric HRQoL instruments. Methods: The planned P-MIC study will collect data on approximately 6100 Australian children and adolescents aged 2–18 years via The Royal Children’s Hospital Melbourne and online survey panels. Participants will complete an initial survey, involving the concurrent collection of a range of pediatric HRQoL instruments, followed by a shorter survey 2–8 weeks later, involving the collection of a subset of instruments from the initial survey. Children aged ≥7 years will be asked to self-report HRQoL. Psychometric performance will be assessed at the instrument, domain, and item level. Conclusions: This paper describes the methodology of the planned P-MIC study, including benefits, limitations, and likely challenges. Evidence from this study will guide the choice of HRQoL measures used in clinical trials, economic evaluation, and other applications.

## 1. Introduction

The increasing recognition of the importance of including the child’s perspective on their health and wellbeing in decision making has resulted in a large increase in the availability and use of pediatric health-related quality of life (HRQoL) instruments [1,2]. Accurate measurement and valuation of HRQoL among children plays a vital role in the development of robust evidence for economic evaluation and population health studies, as well as routine outcomes measurement, clinical decision making and patient choice [1,3].

Economic evaluation frequently incorporates preference-based HRQoL information to produce estimates of Quality Adjusted Life Years (QALYs). QALYs combine HRQoL impacts and survival impacts in a single metric, allowing comparisons across health problems, interventions, and populations. QALYs are based on patients’ HRQoL which is usually captured with validated survey instruments. These instruments comprise a descriptive system covering different HRQoL aspects, which is weighted and scored using QALY weights (or utility values) derived from the stated preferences, usually of the general population for each health state. QALYs inform cost-effectiveness evidence which is formally used in many countries to make decisions about public funding for pharmaceuticals, technologies, and health services. In Australia, bodies such as the Pharmaceutical Benefits Advisory Committee (PBAC) and Medical Services Advisory Committee (MSAC) use evidence about cost effectiveness to inform decisions about the best use of health care resources [4]. These decisions have large cost implications, with approximately $12.6 and $25 billion spent through the Pharmaceutical Benefits Scheme (PBS) and Medicare Benefits Schedule (MBS), respectively, in Australia in 2019–2020 [5,6]. These are federally funded reimbursement schemes ensuring low or no cost to consumers, and similar reimbursement processes are seen around the world.

There is increasing interest in the use of HRQoL information for patient reported outcome measures (PROMs) in clinical practice. In the clinical setting, HRQoL instruments can be used to aid identification of health problems, including unexpected or hidden problems; decision making, including prioritization of health problems or treatments; monitoring of changes in health over time; communication; and patient decision making, including decision making about treatments [7]. PROMs information has been shown to improve HRQoL outcomes and satisfaction with pediatric clinical care [3,8,9]. In this context it is important that the HRQoL instrument is sensitive to change, informative to clinicians and patients, consistent with a patient’s clinical history and allowing for a short collection time. However, there is a paucity of literature on the psychometric properties, feasibility and validity of potential PROM instruments. A 2015 systematic review identified 35 generic paediatric PROMs with none having all their psychometric properties assessed [9]. Eight of the 35 measures identified were preference-based measures [9].

Although there are established methods for measuring and valuing HRQoL in adults [10], considerable challenges arise in applying these methods to children, in particular very young children [11]. For instance, the developmental range of children precludes a ‘one size fits all’ approach and younger children are unable to self-report their HRQoL, requiring proxy completion (usually by parents) [11].

A 2015 systematic review examining the application of preference-based HRQoL instruments in pediatric populations identified nine HRQoL instruments that have been used in pediatric populations (Adolescent Health Utility Measure, Assessment of Quality of Life (AQoL-6D), Child Health Utility (CHU9D), EQ-5D-Y-3L, Health Utilities Index Mark 2 (HUI2) and Mark 3 (HUI3), 16D, 17D, and the Quality of Wellbeing Scale (QWB)) and concluded that more evidence on the performance of these instruments is required [12]. Additionally, a 2021 systematic review examining the psychometric performance of key generic pediatric preference-based instruments (CHU9D, EQ-5D-Y-3L, HUI2, and HUI3) also identified gaps in evidence on the psychometric performance of pediatric HRQoL instruments. Key limitations in the currently available literature include the small sample size of some studies, wide variation in methods, a focus on single instruments, limited coverage and heterogeneity of disease conditions or age groups, meaning that broad synthesis of evidence is challenging [13]. These limitations have resulted in ambiguity regarding which pediatric HRQoL instruments are most appropriate for which context, inhibiting clinicians in instrument use for clinical care, researchers in evidence collection and decision makers in evidence appraisal. No study has previously compared a wide set of generic HRQoL measures accompanied by preference weights concurrently in the same cohort of children (also called a pediatric multi instrument comparison (P-MIC) study).

There is a clear evidence deficit regarding the comparative performance of pediatric HRQoL instruments. Considering the large implications on resource allocation and clinical care, addressing this evidence deficit provides the opportunity to improve value for money across the health care system, resulting in improved population health outcomes for children. This study aims to generate new evidence on the comparative performance of available pediatric HRQoL instruments. This will be done by assessing the validity and reliability of instruments across a range of age and disease groups in a single study. This evidence will improve and inform users’ choice of pediatric HRQoL instruments in Australia and around the world. This paper aims to describe the study methodology of the proposed study.

## 2. Materials and Methods

### 2.1. Study Aims

To compare the performance of a range of pediatric HRQoL instruments in terms of validity, reliability, responsiveness, acceptability, feasibility, and consistency across age and disease groups.

### 2.2. Study Design

This P-MIC study will involve the prospective collection of multiple generic and disease-specific paediatric HRQoL instruments concurrently via two online surveys. The first is a 15–30-min survey, and the second is a five-minute follow-up survey two to eight weeks later. Collecting these paediatric HRQoL instruments concurrently across a range of age and disease groups will allow the relative performance and measurement characteristics of these instruments in children to be assessed. This study was approved by The Royal Children’s Hospital (RCH) Human Research Ethics Committee (HREC/71872/RCHM-2021) on the 20th May 2021 and is prospectively registered with the Australia New Zealand Clinical Trials Registry (ANZCTR) (ACTRN12621000657820).

### 2.3. Timeline

Recruitment for this study began on the 4th of June 2021 and is expected to be completed by January 2022. Data analysis and publication is expected to be ongoing, with main results completed by December 2022.

### 2.4. Participants

Any parent, caregiver, or guardian of a child(ren) aged 2–18 years (inclusive) at study enrolment is eligible to take part. Participants unable to communicate in written English, unable to answer or comprehend the questions or who do not reside in Australia will be excluded from participating. Additional eligibility criteria apply to the disease-specific samples (see Section 2.6.1 for further details).

The study aims to collect data on 6100 Australian children via the recruitment of parents/caregivers. The sample will be split into three groups:Sample one, recruited via hospital: 1000 parents/caregivers will be recruited via The RCH, Melbourne, Australia, allowing data to be collected from children with a wide range of health conditions and severities.Sample two, online panel population sample: 1500 parents/caregivers of children from a general population sample will be recruited via an online survey company, Pureprofile Australia, allowing data to be collected from a large population sample. This sample will be further divided into two groups:○Sample 2a, online panel population sample, normal follow-up: 1300 parent/caregivers will be followed-up in line with the rest of the cohort, two to eight weeks after initial survey.○Sample 2b, online panel population sample, short follow-up: to assess reliability, 200 parents/caregiver will be followed up at a shorter interval, two days after initial survey.Sample three, online panel disease-specific groups sample: 3600 parents/caregivers of children with one of nine health conditions will be recruited via an online survey company, Pureprofile Australia. Disease groups were chosen that had sufficient prevalence to allow for recruitment of groups and with a focus on conditions with larger expected HRQoL decrements [14]. Currently six of the disease groups have been confirmed and include: attention-deficit/hyperactivity Disorder (ADHD), anxiety and/or depression, autism spectrum disorder (ASD), asthma, dental decay, and sleep problems. An additional three disease groups will be selected from the following: diabetes, epilepsy, eating disorder, recurrent abdominal pain, frequent headaches. We aim to collect data on 400 unique children from each of the nine disease groups.

### 2.5. Recruitment

#### 2.5.1. Sample One, Sample Recruited via Hospital (*n* = 1000)

Research Assistants (RAs) will approach parents/caregivers for recruitment from a range of RCH departments, including outpatient clinic waiting rooms, surgical department waiting rooms and the Emergency Department Short Stay Unit (SSU) and provide them with a link to the initial survey. The SSU will provide a list of approved families for RAs to approach on the day when recruiting from this department to ensure appropriate families are approached and to minimize the disruption to care and avoidable distress. Poster adverts with QR codes to the initial survey will be placed in high traffic areas of the hospital. Online advertisements with a link to the initial survey will be placed on hospital telehealth appointments (this advert will appear for any family attending a hospital appointment via telehealth), newsletters, and social media. Recruitment resources will also be utilised to target hospital departments where children likely to be experiencing higher decrements in quality of life, such as children with rare genetic conditions, children with serious sequalae arising across childhood following birth at an extremely low birthweight, and children admitted to the Paediatric Intensive Care Unit (PICU). Potential participants from PICU will be approached by PICU research staff at their pre-operative clinic visit to minimize the recruitment in inappropriate situations (e.g., where the child is unlikely to survive) and to minimize stress to families of children with an unplanned admission. If needed, we may also extend to other areas of the hospital or recruit via other methods such as sharing the advertisement with parent/caregivers from the onsite RCH childcare centre; with participants from a previous study who have indicated they would like to be contacted for future research; sending letters to potential participants from participating hospital departments; and approaching families in the local RCH playground.

#### 2.5.2. Sample Two (Online Panel Population Sample, *n* = 1500) and Sample Three (Online Panel Disease Groups Sample, *n* = 3600)

The recruitment of the online panels will be managed by Pureprofile Australia. All existing members have completed a double opt-in process to join the panel and have agreed to complete online surveys over the course of their membership. They are provided with the opportunity to accept an offer to participate in new surveys. Participants will be randomly selected from this panel to take part in this study if they meet eligibility criteria and fulfil the health status/disease-specific quotas that have been prespecified. Participants will be selected based on quotas for age and sex. As children may have multiple conditions, participants for sample three will be invited to take part on a least fill basis, whereby, children with rarer conditions will be invited to take part for the rarer condition disease group where they will only receive the disease specific measure for this condition even if they report their child has another of the disease specific conditions.

### 2.6. Data Collection

Data will be collected via two online REDCap surveys. Table 1 provides a schedule of which instruments will be collected at which time points for each sample. Appendix A provides a justification for and description of instruments included in the study.

#### 2.6.1. Initial Survey

At the start of the initial survey, parents/caregivers will be asked screening question(s) to confirm their eligibility. Parents/caregivers from every sample will be first asked to confirm they are the parent, caregiver or guardian of a child aged 2–18 years. Parents/caregivers from sample three, the online panel disease groups, will undergo additional screening to ensure they are the parent/caregiver of a child with one of the nine health conditions (Appendix B). Currently six of the nine disease groups and the corresponding screening questions for these groups have been confirmed. Once the additional three disease groups have been confirmed, the corresponding screening questions will be selected for these groups. Following screening, all participants will be required to provide informed consent. The initial survey includes sociodemographic questions, the Strengths and Difficulties Questionnaire (SDQ), EQ Health and Wellbeing Short Version for parents (EQ-HWB-S) followed by several pediatric HRQoL instruments. HRQoL instruments will be blocked into three groups: core, additional and disease-specific. Participants will be randomized to additional instruments to minimize responder burden. Core instruments, which all participants will receive, include: the Global Health Measure, CHU9D, Paediatric Quality of Life Inventory (PedsQL) Core 4.0, EQ-5D-Y, and Toddler and Infant Questionnaire (TANDI). Participants in all samples, where the child is aged five years or older, will receive both the EQ-5D-Y 3L and 5L original version. Participants in sample one, where the child is four years or younger, will receive both the EQ-5D-Y 3L and 5L adapted versions with guidance notes. Participants from samples two and three, where the child is four years or younger, will be randomized to receive either (1) the EQ-5D-Y-3L adapted version with guidance notes and original version or (2) the EQ-5D-Y-5L adapted version with guidance notes and original version. Additionally, participants from samples two and three, the online panel samples, will be randomized to receive one of three additional block(s) of instruments: (1) EQ-5D-5L & HUI2/3, (2) AQol-6D or, (3) Patient-Reported Outcome Measurement Information System 25 (PROMIS-25) Paediatric Profile. Participants from sample three, the online disease group sample, will receive a corresponding disease-specific HRQoL instrument. Currently six of the nine disease groups have been confirmed and the corresponding disease specific instruments for these groups are: PedsQL asthma model (asthma), KIDSCREEEN-27 (ASD), Sleep Disturbance Scale for Children (SDSC) (sleep problems), Child Perceptions Questionnaire (CPQ 11-14) (dental decay), The revised Children’s Anxiety and Depression Scale Short form (RCADS-25) (anxiety and/or depression), and Strengths and Weaknesses of ADHD Symptoms and Normal Behavior Scale (SWAN) (ADHD). Once the additional three disease groups have been confirmed, the corresponding disease specific measure will be selected for these groups. Disease-specific instruments are selected based on the following hierarchical criteria: (1) instrument validated in children, (2) disease-specific QoL instrument, (3) symptom scale or measure, (4) generic QoL instrument commonly used in, or appropriate for use in, children with the condition of interest. Appendix A provides the rationale for choosing each disease specific instrument.

Children aged seven years or older and identified by their parent/caregiver as being able to complete the survey will be invited to complete the HRQoL questions (self-report). If the child is under seven years of age or is unable to complete the survey (e.g., not cognitively able to complete or is undergoing a procedure in hospital) the parent/caregiver will be asked to complete the HRQoL questions on their behalf as proxy. Participants will be given HRQoL instruments to complete that match the child’s age, person reporting (proxy report vs. self-report) and child disease group (if part of the online disease panel) (see Figure 1).

The order in which the HRQoL instruments are presented for completion will be randomized to minimize order and survey fatigue effects. In addition, the EQ-5D-Y-3L, EQ-5D-Y-5L, and, if relevant the EQ-5D-5L, will be presented to participants with another HRQoL instrument separating them, given their similarities. For participants in samples two and three, where the child is four years or younger and is receiving both the EQ-5D-Y (3L or 5L) original and adapted version, the original version will be displayed directly before the adapted version. The order of instruments will be the same for the initial and follow-up survey for each participant. The consent and sociodemographic questions will always appear first in the initial survey.

#### 2.6.2. Follow-Up Survey

A shorter version of the initial survey will be sent out to the majority of participants two to eight weeks after completion of the initial survey. A small proportion (*n* = 200) of participants from sample two, the online population panel sample, will be selected at random and asked to complete the follow-up survey two days post initial survey completion, and this group will form sample 2b. The follow-up survey will comprise the core set of measures described above. Additionally, participants from samples two and three, will receive the same instrument(s) they were randomized to receive in the initial survey. It is expected to take approximately five to ten minutes to complete. For consistency, if the parent/caregiver completed HRQoL questions (proxy report) in the initial survey, they will be asked to again complete the questions on behalf of their child. If a child previously completed the HRQoL questions (self-report) in the initial survey they will be asked to complete them in the follow-up survey. No sociodemographic questions, SDQ or EQ-HWB questions will be included in the follow-up survey. Questions about change in the child’s health status since the first survey will be included in the follow-up survey (See Appendix C).

### 2.7. Participant Duration and Reimbursement

The initial survey will mark the beginning of a participants’ involvement in the study. Participants can enter the study at any point and are not required to be on the same schedule as other participants. The completion of the follow-up survey will mark the end of a participants’ involvement in the study. Participants from sample one, the sample recruited via hospital, will be emailed a $15 online gift voucher once they have completed the follow-up survey to reimburse them for their time. Participants from samples two and three, the online panel samples, will also be reimbursed for their time by Pureprofile Australia, the reimbursement dependent on the questionnaire length, ranging between $3–$15 for a 15–30-min survey. These reimbursed amounts are to compensate participants for some of their time, but are not seen as high enough to unduly coerce.

### 2.8. Sample Size

There is no commonly accepted methodology for calculating sample sizes for studies assessing psychometric properties and hence sample sizes are not often reported [15]. We completed outcome specific sample size calculations where possible to inform recruitment targets for key outcomes, responsiveness, and known-group validity. For all sample size calculations, we focused on the PedsQL measure as it is widely used and has established minimal clinically important difference (MCID) [16]. To ensure enough children were recruited with a change in health status between initial and follow-up surveys, we first calculated the estimated sample size required to assess responsiveness. Using a two-sided paired test with type I error of 1%, it was determined that a sample of 139 would give statistical power of 0.9 to detect the accepted MCID of 4.36 in the total score of the PedsQL self-report [16]. A sensitivity calculation was completed using proxy-report scores and determined that a sample of 190 would give statistical power of 0.9 to detect the accepted MCID of 4.5 in the total score of the PedsQL proxy-report using a two-sided paired test with type I error of 1% [16]. Additionally, we calculated the estimated sample size required to assess known-group validity. Using a one-sided test with type I error of 1%, it was determined that a sample of 186 (93 per group) would give statistical power of 0.9 to detect a difference between healthy children and children with a chronic condition on the total score of the PedsQL proxy report [16]. A sensitivity calculation was completed using the PedsQL self-report scores and it was determined that a sample of 128 (64 per group) would give statistical power of 0.9 to detect a difference between healthy children and children with a chronic condition on the total score of the PedsQL self-report using a one-sided test with type I error of 1% [16]. Additional information on sample size calculations is available in Appendix D. The total sample size of 6100 was estimated from a similar adult study and a review of previous pediatric HRQoL studies [13,17]. Although the overall sample size is greater than the outcome specific sample sizes estimated above, this larger total sample is required to complete further item level analysis such as Item Response Theory (IRT) and factor analysis in addition to subgroup analysis, such as by disease group and child age. Specific sample size calculations have not been performed to support each of these scenarios.

### 2.9. Statistical Analysis

The psychometric properties of the pediatric HRQoL instruments will be analyzed at the overall, domain, dimension, and item levels. They will be tested and compared across the following outcomes: validity, reliability, responsiveness, acceptability and feasibility, and consistency [18]. We will also use Factor Analysis and Modern Test Theory methods such as IRT to understand what the instruments are measuring and the characteristics of individual items. Additionally, we will investigate how the outcomes described above vary by certain factors such as child age, gender, and disease group (including acute versus chronic conditions), family socio-economic status (SES), and presence of anxiety/depression comorbidity. HRQoL instruments will be scored to produce item scores, domain scores and a total utility score where appropriate using published scoring algorithms or value sets to enable further analysis of validity. To minimize missing data we will: (1) work with the online survey panel company to set rules regarding minimum acceptable data, (2) follow up with participants if there is missing survey data at the time of survey completion, and (3) use the functionality of REDCap to require questions to be answered, so that if a question is left blank the survey will prompt the participant to go back and complete the question. Any missing data will be managed by following instrument guidelines.

#### 2.9.1. Validity

Validity is defined as the degree to which the HRQoL instrument measures the construct(s) it purports to measure [18]. Validity will be measured using within-scale analysis (measured using factor analysis), known group differences (measured by descriptively comparing a priori assumptions regarding expected differences between groups), convergent validity (measured by analyzing the correlation of similar constructs from different instruments) and discriminant validity (measured by analyzing whether dimension responses are independent of known groups) [18,19].

#### 2.9.2. Reliability

Reliability is defined as the stability of a participant’s responses on the instrument. Reliability will be assessed using test–retest reliability, measured by agreement on dimension-level responses between the initial survey and the re-test survey two days later [18,19]. This outcome will only be assessed in the *n* = 200 participants from sample 2b.

#### 2.9.3. Responsiveness

Responsiveness is defined as the ability of an instrument to detect change in the construct to be measured [18]. Responsiveness of instruments will be assessed using dimension level responses from children whose proxy respondents reported a change in general health status from the initial survey to the follow-up survey in comparison to those not reporting a change (see Appendix C) [18,19]. This will be assessed to determine the extent to which instruments are responsive to change in general health status.

#### 2.9.4. Acceptability and Feasibility

Acceptability and feasibility will be measured by assessing the time to complete instruments [18,19], and self-reported difficulty (measured by a single, study-designed survey question asking participants to report how difficult the instrument was to complete). Survey missingness or drop-out will also be reported as indicators of acceptability and feasibility, noting the points at which participants prematurely exit the survey.

#### 2.9.5. Consistency

Consistency is defined as the degree to which the summary and dimension specific responses on each instrument are consistent with dimension and summary responses of other similar instruments [18]. Consistency of instruments will be assessed by comparing the consistency of summary and dimension specific responses on each instrument using item-total correlations and Cronbach alpha coefficients [18,19].

#### 2.9.6. Factor Analysis

To assess measurement overlap between instruments we will also use exploratory and confirmatory factor analysis. These techniques identify domains of items measuring similar constructs, and can be used for individual instruments, or to identify domains from a pool of items taken from different instruments.

#### 2.9.7. Item Response Theory

We will use IRT to examine item performance at the dimension and domain level. IRT is a latent scale technique that allows for the assessment of item and model fit, information provided by items within dimensions, and differential item functioning across demographic groups. IRT will be used as an exploratory supplementary tool to add to the understanding of instruments, and the results will be considered in light of the appropriateness of IRT for the instrument (for example considering instrument structure) and the results from other psychometric methods.

### 2.10. Data Security

For sample one, the sample recruited via hospital, only identifying information required for the purposes of follow up will be collected (email and phone number). This identifying information will only be accessed by members of the research team who are required to contact participants for follow up. Data will be captured electronically via REDCap and securely stored in the Murdoch Children’s Research Institute’s (MCRI’s) REDCap database system, which is backed up nightly to a local backup server, with a monthly backup taken to tape and stored offsite. REDCap maintains an audit trail of data create/update/delete events that is accessible to project users who are granted permission to view it. Access to REDCap will be provided via an MCRI user account or (for external collaborators) via a REDCap user account created by the MCRI system administrator. The permissions granted to each user within each REDCap project will be tightly controlled and in accordance with ethics approval.

## 3. Discussion

This study is the first to concurrently collect primary evidence from the same cohort of participants on a broad set of pediatric HRQoL instruments across a diverse sample that is adequately powered to compare the psychometric performance of multiple HRQoL instruments for children.

Potential challenges to this study are the recruitment of a large sample recruited via hospital during the COVID-19 pandemic. Recruitment for this sample is primarily intended to come from face-to-face recruitment which will likely be impacted by any changes to the COVID-19 situation in Victoria, Australia. Additional recruitment strategies outlined in Section 2.5.1 have been planned to help minimize this risk. A limitation of this study is that minimal child level clinical information will be obtained via the survey and thus we will be unable to assess how instruments perform by factors such as disease severity. Responsiveness of HRQoL has traditionally been a difficult area of psychometric performance to assess and our ability to assess responsiveness relies on having between 139 and 190 children who change health status during the period of the study. There are two potential difficulties, firstly in recruiting this number of children who change health status. We have targeted conditions known to have a higher risk of change, nonetheless our ability to recruit adequate numbers of children who change health status is unknown. Secondly, there is no one gold standard method for measuring change in health status via self-reported data. We have relied on several questions aimed at identifying health status change (see Appendix C); however, it is unknown to what extent these questions will adequately identify change. This study can test these factors. Due to the large numbers of children to be recruited and the desire to seek children with a variety of conditions, it is unknown how many will be recruited with each condition in the sample recruited via hospital. Specific strategies to target more seriously ill children will be employed, but it is unknown what this balance will be prior to understanding how recruitment success progresses. Another limitation of the study is the decision around the length of follow-up. For consistency, a single follow-up period will be selected for all samples, excepting the test–retest sample (sample 2b). A period that is too short will perhaps not give some groups of children enough time to change their health status and this will be different for each condition. A period that is too long will place pressure on completion rates. There is likely to be no one right answer across all patient groups so a consistent compromise will be sought. In addition, once a follow up period is specified, it is unlikely that all participants will complete in the exact same amount of time. Variation by a couple of days/weeks either side of the set follow up time is expected which creates some variation in the data. However, this variation will be recorded to allow for its impact to be tested in analyses.

With limited resources and increasingly costly medications and technologies potentially available for children, it is important that decisions about the allocation of public funding for medications and technologies are informed by best possible evidence, including cost-effectiveness evidence. The accurate measurement of HRQoL is key to ensuring that utility as an input to cost-utility analysis is accurate, but currently there is a lack of sound evidence on the psychometric performance of pediatric HRQoL instruments. This lack of evidence is impacting the ability of decision makers to make informed choices about pediatric interventions. The data collected from this study provides an additional opportunity to psychometrically assess potential instruments for use as a routine clinical PROM in paediatric clinic settings, thereby filling this—and other gaps in the literature—including the lack of comparative data across instruments and child age and conditions. Future work from this study could look at international extensions and the development of psychometric testing protocols. Our work will allow for clear comparison of evidence across instruments [13] and will be of national and international significance.

## Figures and Tables

**Figure 1 children-08-00714-f001:**
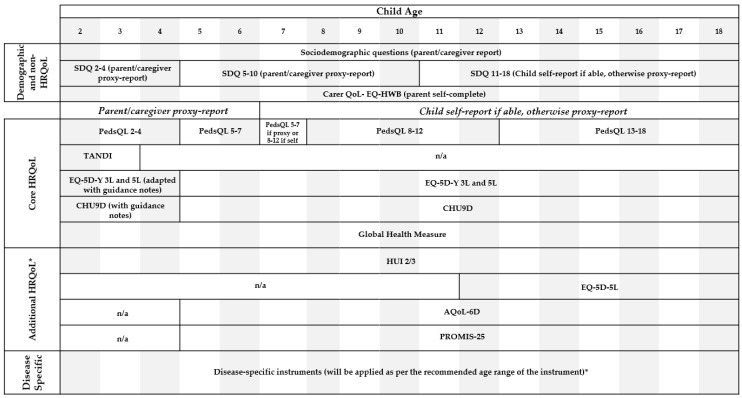
Summary of instruments by age. * Participant will only receive if allocated instrument based on disease group and/or randomization to receive additional instrument. Abbreviations: HRQoL health-related quality of life, SDQ Strengths and Difficulties Questionnaire, EQ-HWB EQ Health and Wellbeing Short Version, PedsQL Paediatric Quality of Life Inventory, TANDI Toddler and Infant Questionnaire, EQ-5D-Y EQ-5D Youth, CHU9D Child Health Utility, HUI2/3 Health Utilities Index Mark 2/3, EQ-5D-5L, AQoL-6D Assessment of Quality of Life, PROMIS-25 Patient-Reported Outcome Measurement Information System 25.

**Table 1 children-08-00714-t001:** Summary of instruments by study sample.

Instrument	Sample 1, Sample Recruited via Hospital		Sample 2, Online Panel Population Sample		Sample 3, Online Disease Group Sample	
	Initial	Follow-Up	Initial	Follow-Up	Initial	Follow-Up
**Demographic and non-HRQoL instruments**						
Sociodemographic questions	X		X		X	
SDQ	X		X		X	
EQ-HWB	X		X		X	
**Core HRQoL instruments**						
PedsQL	X	X	X	X	X	X
TANDI (if <=3 years)	X	X	X	X	X	X
EQ-5D-Y 3L & 5L original (if >=5 years)	X	X	X	X	X	X
EQ-5D-Y 3L & 5L adapted (if <=4 years)	X	X				
EQ-5D-Y 3L original and adapted or EQ-5D-Y 5L original and adapted (if <=4 years)			X *	X *	X *	X *
CHU9D	X	X	X	X	X	X
Global Health Measure (single item)	X	X	X	X	X	X
**Additional HRQoL instrument blocks**						
HUI 2/3 and EQ-5D-5L (>11 years)			X *	X *	X *	X *
AQoL-6D (>4 years)			X *	X *	X *	X *
PROMIS-25 (>4 years)			X *	X *	X *	X *
**Disease specific instruments**						
Disease specific instruments (as per the recommended age range of the instrument)					X *	

X- indicates the instrument will be collected from the sample/time point. * Participant will only receive, if allocated, instrument based on disease group, and/or randomization to receive additional instrument, and/or randomization to receive EQ-5D-Y 3L original and adapted or EQ-5D-Y 5L original and adapted. Abbreviations: HRQoL health-related quality of life, SDQ Strengths and Difficulties Questionnaire, EQ-HWB EQ Health and Wellbeing Short Version, PedsQL Paediatric Quality of Life Inventory, TANDI Toddler and Infant Questionnaire, EQ-5D-Y EQ-5D Youth, CHU9D Child Health Utility, HUI2/3 Health Utilities Index Mark 2/3, EQ-5D-5L, AQoL-6D Assessment of Quality of Life, PROMIS-25 Patient-Reported Outcome Measurement Information System 25.

## Data Availability

Not applicable.

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
