# Peer review of "Psychometric Performance of HRQoL Measures: An Australian Paediatric Multi-Instrument Comparison Study Protocol (P-MIC)"

_children, 2021, doi:10.3390/children8080714_

Round 1

Reviewer 1 Report

The protocol entitled ‘Psychometric performance of HRQoL measures: an Australian pediatric multi-instrument comparison study protocol (P-MIC)’ has been well planned and the research team has considered various potential hurdles which will invariably arise with large-scale studies like this. As the paper is a planned protocol, it is to be expected that the description of the analysis of data is a little sparse. I would advise the researchers to consider the potential issues which may arise with IRT analysis if data is not complete and if they would have sufficient sample size to merit differential item functioning, for example, across specific disease groups (the authors do mention the hoped for sample sizes in these groups but if you are going to set about detecting for potential DIF, then best you ensure you get as many ppts in these groups as possible). There are of course various IRT models which would work for the different scales: polytomous scores, binary scores etc. Depending on how the original psychometric analyses of the various health-related measures was conducted, IRT may or may not reveal some surprises. Bear in mind that the various measures may be a mixed bag in terms of uni- and multidimensional underlying constructs. Designing the study necessarily needs to consider how the data will be analysed. Just two minor points:

  • lines 125-129: How will you control for VPN use? Although unlikely to be the case very often, setting up an Australian VPN can still allow ppts to complete the online survey from outside the country. Perhaps you are going to use a national ID of sorts? If anonymity is to be maintained however, how will you control for this? Also, will there be a short comprehension reading test to ensure that ppts can indeed understand the items? These items are usually written at an 11-year-old level so this will likely not be an issue, but it is worth considering.
  • Lines 157-158: What about cross-disease burden? This is a difficult issue to consider when designing the study. Will there be an attempt to deal with cases where children (or their parents) report multiple morbidities? E.g., reporting headaches and recurrent abdominal pain may occur more frequently in tandem with sleep disturbances. The assumption is that if ppts pass the screening questions on several of the additional screening questions for sample three that they will simply complete all the instruments that are appropriate. The choice of the nine diseases (six of which have been confirmed) do seem to be fairly differentiated so this issue is not a significant one.

Reviewer 2 Report

The authors of this submission planned a pediatric multi-instrument comparison study with the aim of comparing the psychometric performance and measurement characteristics of pediatric health-related quality of life instruments.

This is a refined protocol. The Introduction section provides a clear and useful background to understand the research. Study objectives are clear. Methods for recruitment and for data collection are well described. I am familiar with REDcap, which surely is an adequate tool. The table and the figure provide an accurate summary of the project and are presented in a precise fashion. The Discussion section validly completes the manuscript. I appreciated the fact that the appendices are very detailed.

Overall, I have no concerns about this submission.

Author Response

We thank the reviewer for taking the time to review the manuscript and for their kind comments.